# Nanobodies against *C. difficile* TcdA and TcdB reveal unexpected neutralizing epitopes and provide a toolkit for toxin quantitation *in vivo*

**Shannon L. Kordus**[1☉], **Heather K. Kroh**[1☉], **Rubén Cano Rodríguez**[1], **Rebecca A. Shrem**[2], **F. Christopher Peritore-Galve**[1], **John A. Shupe**[1], **Brian E. Wadzinski**[2], **D. Borden Lacy**[1,3], **Benjamin W. Spiller**[1,2]*

**1** Department of Pathology, Microbiology, and Immunology, Vanderbilt University Medical Center, Nashville, Tennessee, United States of America, **2** Department of Pharmacology, Vanderbilt University, Nashville, Tennessee, United States of America, **3** Department of Veterans Affairs, Tennessee Valley Healthcare System, Nashville, Tennessee, United States of America

☉ These authors contributed equally to this work.
* benjamin.spiller@vanderbilt.edu

**Data Availability Statement:** All relevant data are within the paper. Sequences to clones developed in this work can be found in the supplemental figures.

## Abstract

*Clostridioides difficile* is a leading cause of antibiotic-associated diarrhea and nosocomial infection in the United States. The symptoms of *C. difficile* infection (CDI) are associated with the production of two homologous protein toxins, TcdA and TcdB. The toxins are considered *bona fide* targets for clinical diagnosis as well as the development of novel prevention and therapeutic strategies. While there are extensive studies that document these efforts, there are several gaps in knowledge that could benefit from the creation of new research tools. First, we now appreciate that while TcdA sequences are conserved, TcdB sequences can vary across the span of circulating clinical isolates. An understanding of the TcdA and TcdB epitopes that drive broadly neutralizing antibody responses could advance the effort to identify safe and effective toxin-protein chimeras and fragments for vaccine development. Further, an understanding of TcdA and TcdB concentration changes *in vivo* can guide research into how host and microbiome-focused interventions affect the virulence potential of *C. difficile*. We have developed a panel of alpaca-derived nanobodies that bind specific structural and functional domains of TcdA and TcdB. We note that many of the potent neutralizers of TcdA bind epitopes within the delivery domain, a finding that could reflect roles of the delivery domain in receptor binding and/or the conserved role of pore-formation in the delivery of the toxin enzyme domains to the cytosol. In contrast, neutralizing epitopes for TcdB were found in multiple domains. The nanobodies were also used for the creation of sandwich ELISA assays that allow for quantitation of TcdA and/or TcdB *in vitro* and in the cecal and fecal contents of infected mice. We anticipate these reagents and assays will allow researchers to monitor the dynamics of TcdA and TcdB production over time, and the impact of various experimental interventions on toxin production *in vivo*.

**Funding:** This work was supported by GM139303 to SLK, AI095755, AI174999, and BX002943 to DBL, and AI174999 to BWS. The funders had no role in study design, data collection and analysis, decision to publish, or preparation of the manuscript.

**Competing interests:** BWS and BEW are founders and principals at Turkey Creek Biotechnology LLC (Waverly TN). Turkey Creek Biotechnology performed the alpaca immunizations and blood draws but was not involved in subsequent work.

## Author summary

*C. difficile* (*C. diff*) is a leading cause of diarrhea and is recognized as an urgent threat by the Centers for Disease Control. Disease symptoms are caused by two large, similar, protein toxins, TcdA and TcdB. These toxins are drug targets and are also important for diagnosis. Despite their clear importance, the understanding of how to neutralize toxin activity is incomplete, and there are no freely available tools to quantify toxin concentration in research studies. To address these issues, we have developed nanobodies that bind and neutralize TcdA and TcdB and have also used these nanobodies to develop quantitative assays for TcdA and TcdB detection. Neutralization studies led us to discover that many of the potent neutralizers of TcdA bind epitopes within the delivery domain. This finding suggests either a role for the delivery domain in receptor binding or that the nanobodies block pore-formation and thereby inhibit delivery of the toxin enzyme domains to the cytosol. The availability of nanobody assays that can differentiate the quantities of TcdA from TcdB should permit a better understanding of toxin-specific effects and how toxin levels change over the course of infection.

## Introduction

*Clostridioides difficile* is the causative agent of *C. difficile* infection (CDI), a disease with symptoms ranging from mild diarrhea to more life-threatening conditions such as pseudomembranous colitis and toxic megacolon [1]. *C. difficile* has been classified as a "threat-level urgent" pathogen by the US Centers for Disease Control due to high levels of morbidity and mortality [2,3]. Treatment options are limited, and vaccine efforts have not succeeded, motivating ongoing research into novel prevention and therapeutic strategies.

Symptoms of CDI are associated with the production of up to three toxins, TcdA, TcdB, and CDT [4]. TcdA and TcdB are considered the main virulence factors and are large (308 and 270 kDa, respectively) glucosylating toxins which inhibit Rho-family GTPases. Multiple large scale vaccine trials focused on the use of toxoid antigens have shown promise in pre-clinical models but have failed to meet primary clinical endpoints in people [5,6]. Among multiple avenues for optimization, there is a need to better understand the toxin sequences and structures that promote broadly neutralizing antibody responses. This is especially true for TcdB, as it is now appreciated that TcdB sequences can vary, with 5 or more sub-types (TcdB1-B5) prevalent among circulating clinical strains [7,8].

TcdA and TcdB share ~47% sequence identity and have four domains: the N-terminal glucosyltransferase domain (GTD), the autoprocessing domain (APD), the delivery domain (DD), and the combined repetitive oligopeptides (CROPs) domain [4]. TcdA and TcdB intoxicate cells via a multistep process: receptor binding and endocytosis, pH-dependent pore formation, translocation of the GTD and APD across the endosomal membrane, autoprocessing and GTD release, and finally, GTD-mediated glucosylation of host GTPases. In cell culture models, the glucosyltransferase activity causes cell rounding, a loss of tight junction formation, and apoptotic cell death, and studies performed in small animal models of infection point to a clear role for the glucosyltransferase activity in pathogenesis [9–12].

TcdA and TcdB can bind to multiple receptors, but not all receptor binding sites have been defined [13]. Historically, receptor binding has been associated with the C-terminal CROPS domain. The TcdA CROPS has 7 repetitive sequence blocks (R1-R7) while the TcdB CROPS has 4 (R1-R4), and this repetition has been hypothesized to contribute to immunodominance [9,14]. The repetitive sequence blocks engage glycans with low affinity, but these

could promote high avidity interactions if multiple glycans are engaged simultaneously [15]. More recent studies have identified receptor interactions outside of the CROPS domain. For example, TcdA can engage sulfated glycosaminoglycans in a CROPS-independent interaction with the sulfate group [16]. TcdB has been reported to bind four protein receptor classes: chondroitin sulfate proteoglycan 4 (CSPG4); Frizzled (FZD) 1, FZD2, and FZD7; Nectin3; and TFP1 [17–20]. The interactions can vary with TcdB subtype, but arguably all bind in a CROPS-independent interaction. (The interaction with CSPG4 occurs at the 'hinge' where the CROPS moves relative to the other toxin domains and involves some of the N-terminal residues of the CROPS [21].) Like the variation in receptor binding with TcdB subtype, the neutralization efficacy of different monoclonal antibodies can differ depending on the TcdB subtype [22].

In addition to the need to define broadly neutralizing epitopes, we have found a need to assay and quantify the concentrations of TcdA and TcdB in our animal model of infection. While ELISAs are commercially available for the detection of TcdA and TcdB in human stool and are considered a key component of CDI diagnostics in many clinical microbiology laboratories, they have limited sensitivity and are considered non-quantitative for research purposes. Many labs make use of Vero cell rounding assays and a proprietary reagent that neutralizes TcdA/TcdB-induced rounding, but the reagent cannot differentiate between TcdA and TcdB. Some labs have access to a real time cellular impedance assay, which is sensitive and quantitative but is expensive and also not readily able to differentiate between TcdA and TcdB [23,24].

Nanobodies are small (12–14 kDa), antigen-binding proteins derived from the heavy-chain only antibodies found in camelids [25]. The nanobody paratope consists of three complementarity determining regions (CDR1/2/3) and can also include framework residues. Of these, the nanobody CDR3 is often longer than what is found in typical mouse or human antibodies and is often the main driver of antigen recognition and specificity. We pursued a nanobody development project with the goal of identifying neutralizing toxin epitopes and reagents that could also be used for research-related toxin quantitation in the feces and cecal contents of mice. Two previous TcdA/TcdB nanobody development efforts have been reported: one that immunized llamas with a portion of the CROPS domain [26] and one in alpacas that used TcdA and TcdB holotoxins with point mutations in the GTD that rendered the toxins glucosyltransferase-deficient [9,26–28]. In our study, we immunized alpacas with a different but similar glucosyltransferase-deficient variant of TcdA, TcdA D285N/D287N, hereafter referred to as $TcdA_{GTX}$ [12]. As prior work had indicated residual toxicity associated with glucosyltransferase-deficient variants of TcdB, we used a TcdB variant with a L1106K mutation that renders the toxin deficient for pore-formation [29]. This mutant has been characterized and shown to be non-toxic in mice [30,31]. In addition to different vaccination regimens, another key difference in our study was that we screened for nanobodies that could bind discrete domains within the holotoxin. With these clones, we sought to expand the reagents available to study the mechanisms of intoxication and disease caused by TcdA and TcdB.

## Results

### Nanobody development

As outlined in Fig 1, two alpacas were immunized with either $TcdA_{GTX}$ (a glucosyltransferase-deficient mutant of TcdA) or TcdB L1106K, a non-toxic version of a TcdB1 sequence derived from the VPI10463 strain. 'Miracle' was immunized eight times with $TcdA_{GTX}$, and 'CaLee' was similarly immunized with TcdB L1106K. Following immunization, blood was drawn, peripheral blood mononuclear cells (PBMCs) were isolated, total RNA was purified,

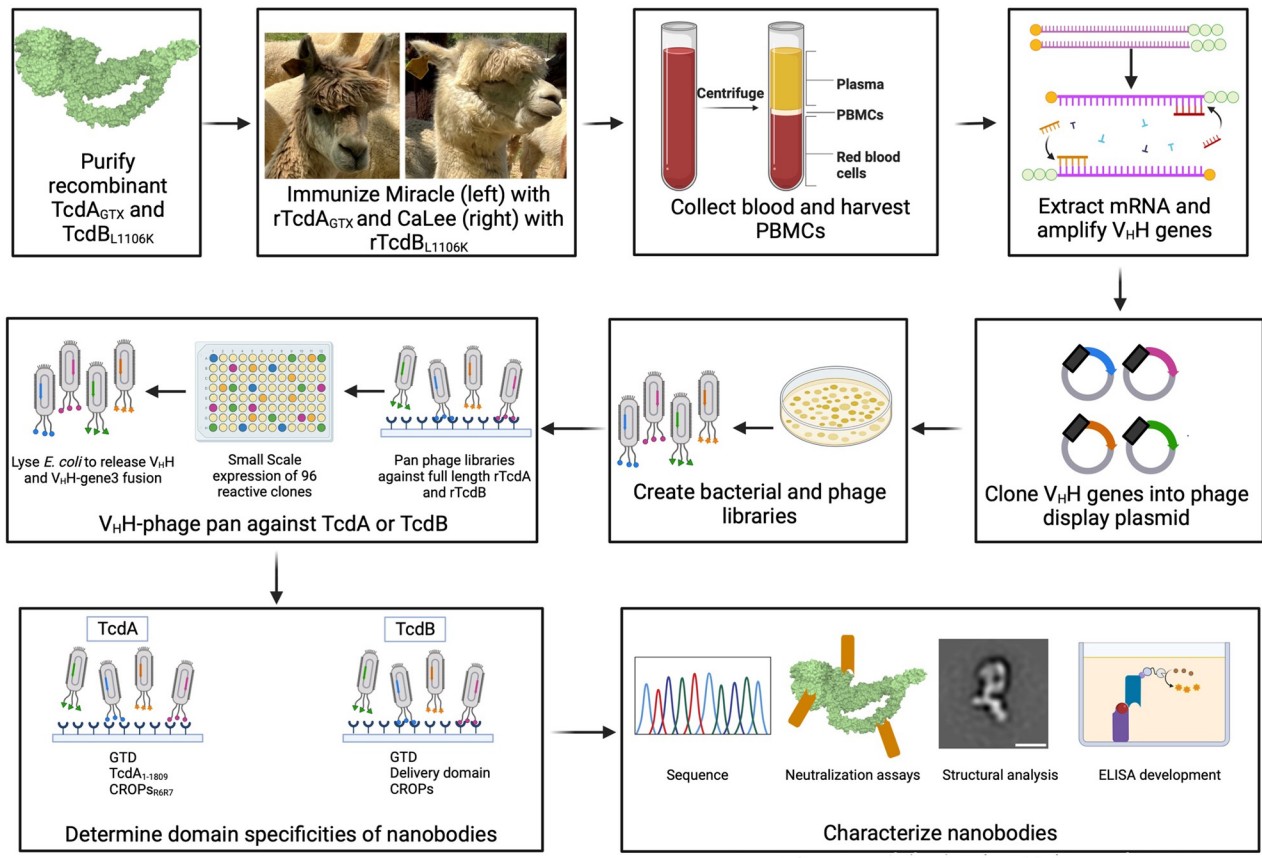

**Fig 1. A summary of the workflow for isolating alpaca-derived nanobodies against TcdA and TcdB.** Created with Biorender.com.

cDNA was made, and phage display libraries were produced. These libraries are not cell type specific, include representation from all circulating B-cells, and are not B-cell lineage specific. A single round of panning using TcdA$_{GTX}$ and TcdB L1106K was done using each library, and reactive phage were recovered into *E. coli* and stored as glycerol stocks. From these libraries, deep-well 96 well plates were inoculated with single clones recovered from the pans (three plates from the TcdB L1106K pan and two from the TcdA$_{GTX}$ pan), and protein expression was induced. Lysates from these clones were used in ELISA-like assays against either purified full-length toxins or isolated domains. In total, each TcdA supernatant was screened against full-length TcdA, the TcdA glucosyltransferase domain (GTD), a GTD-APD-DD construct lacking the CROPS domain (TcdA$_{1-1809}$) or repeats 6 and 7 of the TcdA CROPS (TcdA$_{R6R7}$). Each TcdB supernatant was screened against full-length TcdB1, the TcdB2 glucosyltransferase domain (GTD), the TcdB1 delivery domain (TcdB$_{842-1834}$) or the TcdB1 CROPS domain (TcdB$_{1827-2366}$). Clones encoding nanobodies that recognized full length TcdA and TcdA$_{1-1809}$ but failed to recognize TcdA GTD were tentatively assigned as APD-DD binders.

All clones (five-96 well plates) were sequenced, and cladograms were assembled based on the protein sequences (S1 Fig and S1 Table). The antigenic sequences were well-distributed across the length of the four-domain toxins (Table 1). TcdA$_{GTX}$ immunization resulted in 69 unique clones: 14 against GTD, 30 against APD-DD, and 19 against CROPS R6R7. TcdB

**Table 1. Domain recognition of TcdA and TcdB nanobody clones.**

|  | Total sequences | Non-redundant |
|---|---|---|
| **TcdA** | 176 | 69 |
| *Domain specificity* |  |  |
| GTD | 50 | 14 |
| APD-DD | 63 | 30 |
| CROPs | 54 | 19 |
| *n.d* | 9 | 6 |
| **TcdB** | 267 | 126 |
| *Domain specificity* |  |  |
| GTD | 106 | 43 |
| DD | 50 | 35 |
| CROPs | 98 | 41 |
| *n.d* | 13 | 7 |

n.d. = not determined

L1106K immunization resulted in 126 unique clones: 43 against GTD, 35 against DD, and 41 against CROPS. Sequence analysis identified substantial redundancy in the TcdA panel, such that only 69 of 176 recovered clones were unique, with some clones being particularly common. The TcdB panel was somewhat more diverse, with 126 of 267 clones being unique (Table 1). Each group had a few relatively dense clades (S1 Fig) with each group having a single very successful clone. For example, the A1A6 sequence (recovered from the TcdA immunized animal) was represented 30 times, while the B1C10 sequence (recovered from the TcdB immunized animal) was represented 26 times. Additional sequence analysis to map the likely germline origin of these clones (Table 2) revealed somewhat limited V gene usage, with a single V gene responsible for ~75% of the clones identified from each animal. IGHV 3–3 was the inferred germline parent of 71% of TcdA clones, and IGHV 3S53 was the inferred germline parent of 79.4% of the TcdB clones. Inferred D gene usage was variable, but inferred J gene usage also showed a small number of dominant gene segments. 98.9% of TcdA clones use IGHJ4, 44.2% of TcdB clones use IGHJ6, and an additional 37.5% of TcdB clones use IGHJ4. As annotated in IMGT, there are 17 V regions associated with VHH antibodies in alpacas, 8 D regions, and 7 J regions [32], highlighting the constrained gene usage and the importance of CDR3, which is often critical for antigen recognition.

In general, sequence redundancy was used to guide clone selection for further study. For instance, the screen for anti-GTD nanobodies in the TcdA library yielded 30 identical clones. A representative of this clade, A1A6, was selected for further study. Of the 25 TcdA clones selected for study, 16 were recovered multiple times (S1A Fig). Similarly, 26 identical GTD directed clones were found within the TcdB panel and 11 of the 17 TcdB clones selected for additional study were identified more than once (S1B Fig). Following promising results in neutralization studies, described below, the TcdA anti-APD-DD panel was expanded such that a total of 18 clones in this group were characterized. The highlighted antibodies (S1 Fig) were all expressed and purified to homogeneity.

## Toxin neutralization

Toxin neutralization experiments were performed using purified toxin and purified nanobodies in a cell viability assay. Two cell lines were screened for each toxin to address potential

**Table 2. Inferred gene use of TcdA and TcdB targeting clones.**

| TcdA (total clones = 176) | | | TcdB (total clones = 267) | | |
|---|---|---|---|---|---|
| V/D/J Gene | # Clones | Percent | V/D/J Gene | # Clones | Percent |
| IGHV 3–3 | 125 | 71.0 | IGHV 3–3 | 5 | 1.9 |
| IGHV 3S53 | 15 | 8.5 | IGHV 3S6 | 1 | 0.4 |
| IGHV 3S61 | 3 | 1.7 | IGHV 3S9 | 1 | 0.4 |
| IGHV 3S66 | 31 | 17.6 | IGHV 3S42 | 1 | 0.4 |
| IGHV 3S65/3S66 | 2 | 1.1 | IGHV 3S53 | 212 | 79.4 |
| | | | IGHV 3S54 | 2 | 0.7 |
| IGHD 1 | 9 | 5.1 | IGHV 3S55 | 2 | 0.7 |
| IGHD 2 | 29 | 16.5 | IGHV 3S56 | 3 | 1.1 |
| IGHD 3 | 36 | 20.5 | IGHV 3S60 | 6 | 2.2 |
| IGHD 4 | 32 | 18.2 | IGHV 3S61 | 3 | 1.1 |
| IGHD 5 | 26 | 14.8 | IGHV 3S65 | 16 | 6.0 |
| IGHD 6 | 1 | 0.6 | IGHV 3S67 | 1 | 0.4 |
| IGHD 7 | 7 | 4.0 | Other | 14 | 5.2 |
| *n.d.* | 36 | 20.5 | *IGHV 3S10/3S6/3S9* | *1* | *0.4* |
| | | | *IGHV 3S30/3S31* | *3* | *1.1* |
| IGHJ 4 | 174 | 98.9 | *IGHV 3S36/3S53* | *1* | *0.4* |
| IGHJ 6 | 1 | 0.6 | *IGHV 3S39/3S42* | *1* | *0.4* |
| IGHJ 7 | 1 | 0.6 | *IGHV 3S53/3S54/3S56/3S57* | *8* | *3.0* |
| *\*n.d. No result in IGMT analysis* | | | IGHD 1 | 26 | 9.7 |
| | | | IGHD 2 | 28 | 10.5 |
| | | | IGHD 3 | 41 | 15.4 |
| | | | IGHD 4 | 16 | 6.0 |
| | | | IGHD 5 | 66 | 24.7 |
| | | | IGHD 6 | 34 | 12.7 |
| | | | IGHD 7 | 8 | 3.0 |
| | | | IGHD 8 | 2 | 0.7 |
| | | | *n.d.* | 46 | 17.2 |
| | | | IGHJ 2 | 3 | 1.1 |
| | | | IGHJ 3 | 11 | 4.1 |
| | | | IGHJ 4 | 100 | 37.5 |
| | | | IGHJ 4/6 | 14 | 5.2 |
| | | | IGHJ 6 | 118 | 44.2 |
| | | | IGHJ 7 | 20 | 7.5 |
| | | | *no rearrangement* | 1 | 0.4 |

differences in surface receptor repertoires present on the cells. In the case of TcdA where the relevant receptor repertoire remains undefined, nanobody-dependent neutralization was assessed using T84 and/or Vero cells. For TcdB, Caco-2 and/or Vero cells were used. It is known that the TcdB1 receptor, CSPG4, is present on Vero cells but not Caco-2 cells and that the neutralization properties of antibodies can vary depending on cell type [33]. Neutralization data are summarized in Table 3, and viability curves are shown in S2 Fig. Despite the functionally similar domains, the neutralization data suggest significant differences in how the two toxins can be inhibited. Within TcdA clones, strong neutralizing activity was seen only against the putative APD-DD targeted clones, with 15 of 18 nanobodies able to neutralize and 5 of these clones showing sub-nanomolar neutralization. The GTD-targeted clone, A1A6, showed no neutralizing activity. In contrast, all but one of the GTD-targeted TcdB nanobodies were

**Table 3. Neutralization potency of TcdA and TcdB nanobodies.**

| Nanobody | TcdA Domain | EC50 (nM) | | Nanobody | TcdB Domain | EC50 (nM) | |
|---|---|---|---|---|---|---|---|
| | | T84 | Vero | | | Caco-2 | Vero |
| A1A6 | GTD | n.d. | n.t. | B1C10 | GTD | n.t. | 2.2 |
| | | | | B1E7 | GTD | n.t. | 10 |
| A2F12 | APD-DD | 1.1 | < 1 | B0D11 | GTD | 110.9 | 176 |
| A1D8 | APD-DD | 2.1 | < 1 | B0B11 | GTD | 160.7 | 350 |
| A2C2 | APD-DD | 2.4 | < 1 | B0B7 | GTD | n.d. | < 1 |
| A2G6 | APD-DD | 9.2 | < 1 | B2C5 | GTD | n.t. | < 1 |
| A1C11 | APD-DD | 10.4 | < 1 | B2C11 | GTD | n.t. | < 1 |
| A1D1 | APD-DD | n.t. | 1.5 | B0D3 | GTD | n.d. | n.d. |
| A1G6 | APD-DD | n.t. | 6.4 | | | | |
| A1H5 | APD-DD | n.t. | 15.7 | B0C10 | DD | < 1 | 1.3 |
| A2B5 | APD-DD | n.t. | 11.1 | B1C11 | DD | n.t. | 10 |
| A1C4 | APD-DD | 60.3 | 2.5 | B0A9 | DD | 1.3 | 12.9 |
| A2A6 | APD-DD | 91 | 1 | B0E2 | DD | < 1 | 28 |
| A2H9 | APD-DD | 105.3 | 8.7 | B0A12 | DD | n.d. | 141 |
| A1A3 | APD-DD | 135.5 | 3.7 | B0D10 | DD | < 1 | < 1 |
| A2H4 | APD-DD | 838.3 | 40.9 | | | | |
| A1G4 | APD-DD | 843.7 | 8.5 | B2F11 | CROPs | n.t. | 15.6 |
| A2G5 | APD-DD | > 10 μM | n.t. | B1A11 | CROPs | n.t. | 38.4 |
| A1F4 | APD-DD | > 10 μM | n.d. | n.d. = not detected within tested range | | | |
| A1C3 | APD-DD | n.d. | n.t. | n.t. = not tested | | | |
| | | | | | | TcdA | TcdB |
| A2F10 | CROPs-R6R7 | 455.2 | n.t. | **Tested for neutralization** | | 25 | 16 |
| A2B10 | CROPs-R6R7 | 677.5 | n.t. | **Neutralizers (EC$_{50}$ < 1 μM)** | | 17 | 15 |
| A1C1 | CROPs-R6R7 | > 10 μM | n.t. | **Specificity of neutralizers** | | | |
| A1H1 | CROPs-R6R7 | > 10 μM | n.t. | *GTD* | | 0 | 7 |
| A2A8 | CROPs-R6R7 | > 10 μM | n.t. | *APD-DD or DD* | | 15 | 6 |
| A2G1 | CROPs-R6R7 | n.d. | n.t. | *CROPs* | | 2 | 2 |

neutralizing, with multiple potent neutralizers within the group. Similarly, only weak neutralizers were identified against the TcdA CROPS R6R7 region despite potent neutralizers being found against the TcdB CROPS.

## Epitope identification

Negative stain electron microscopy (EM) with single particle averaging is a method in which dispersed proteins or complexes are coated with a high contrast stain and imaged with an electron microscope. Many orientations are imaged simultaneously, and individual particles are then aligned and grouped by their shape and orientation. The signal enhancement obtained by averaging particles within a class or group can often provide an indication of two-dimensional shape and size. We performed negative stain EM with single particle averaging for a subset of nanobodies to confirm binding to toxin and to locate the binding site of each nanobody (Fig 2). We were particularly interested in the nanobodies where we had deduced binding to the TcdA APD-DD. The A1D1, A2B5, A1D8, and A2H9 nanobodies all bind the DD, and all neutralize (Figs 2 and S2 and Table 3). A1C3, a non-neutralizing TcdA clone, binds near the interface of the GTD, APD, and DD (Fig 2). As with TcdA, we focused on locating the binding site of a subset of TcdB nanobodies. B0E2 and B0D10 bound the DD, and both neutralized (Figs 2

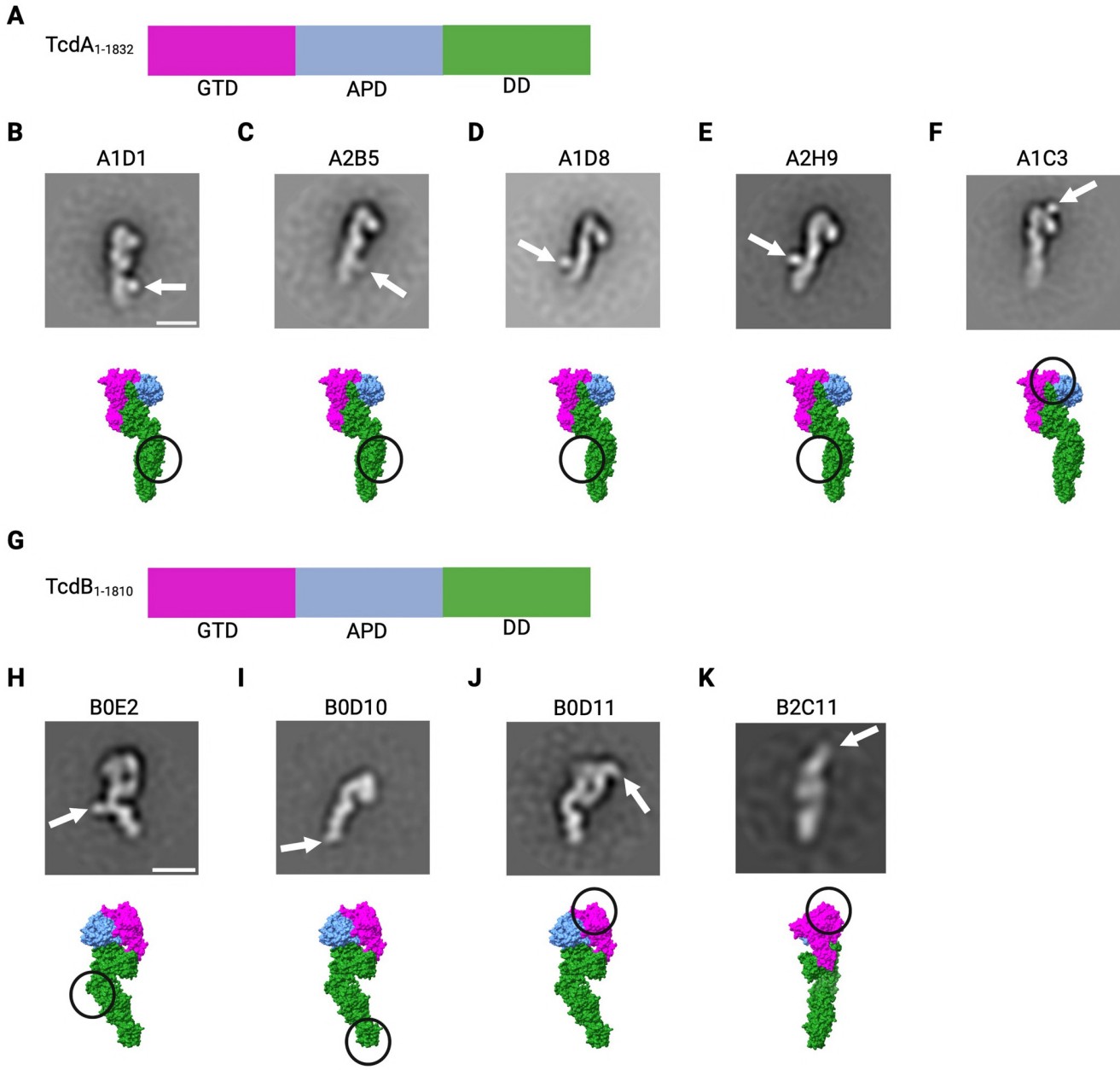

**Fig 2. Negative stain EM of TcdA- and TcdB-nanobody (Nb) complexes.** A). TcdA$_{1-1832}$ domain organization: Magenta, GTD (glucosyltransferase domain); Blue, APD (autoprocessing domain); Green, DD (delivery domain). 2D class averages determined from TcdA$_{1-1832}$ with 2-fold molar excess of nanobody. White arrow indicates location of Nb. B) A1D1, C) A2B5, D) A1D8, E) A2H9, or F) A1C3. Below are space-filling structures using domain colors found in A) (PDB: 4R04) with the Nb binding locations from the 2D averages circled G). TcdB$_{1-1810}$ domain organization colored as in A). 2D class averages determined from TcdB$_{1-1810}$ with 2-fold molar excess of Nb. White arrow indicates location of Nb. H) B0E2, I) B0D10, J) B0D11, or K) B2C11. Below are space filling structures using domain colors found in A) (PDB: 6OQ5) with the Nb binding locations from the 2D averages circled. Scale bar: 100 Å. Image created with BioRender.com license EY25IG35BM.

and S2 and Table 3). Both B0D11 and B2C11 bound the GTD, and both could neutralize (Figs 2 and S2 and Table 3).

## ELISA Development

With this understanding of where nanobodies were binding, we chose to develop sandwich ELISA-based quantification assays for TcdA and TcdB. In all assays, plates were coated with a capture nanobody, and the detection nanobody was biotinylated at a C-terminal Avi-tag to enable use of a streptavidin-horseradish peroxidase (HRP) for the readout. For TcdA, a CROPS-targeted capture nanobody, A2B10, was used in conjunction with the GTD-targeted detection nanobody, A1A6 (Fig 3 panels A-F). This pairing was effective at detecting TcdA when added to buffer (Fig 3A) or culture supernatant (from a toxin-deleted strain), showing a limit of detection (LOD) of 0.6 ng/mL (Fig 3B). The specificity of the nanobody pair for TcdA was evaluated in bacterial cultures where TcdA, TcdB, or both TcdA and TcdB had been deleted. The assay was only effective (with signal above the LOD) in the strain expressing TcdA (M7404 *tcdB*::*ermB*) (Fig 3C). We noted a wide range of toxin levels across the three biological replicates, consistent with prior reports documenting the complex variables affecting toxin production [34,35]. We explored this for three strains of *C. difficile*, where three independent colonies were grown and sub-cultured for 24 hours. The supernatants were filtered and tested in the sandwich ELISA, revealing TcdA concentration ranges of 65–124 ng/mL in M7404, 145–290 ng/mL in R20291, and 326–1380 ng/mL in VPI10463 (Fig 3D). Next, we tested our ability to detect purified rTcdA that was spiked into resuspended mouse feces or mouse cecal content. The LOD for the A2B10/A1A6 pairing was 12 ng/mL in these complex mixtures, 20-fold less sensitive than what was observed in PBS or bacterial media (Fig 3B, 3E and 3F). To address this limitation, we then developed an assay using A1D8 and A1C3 as capture and detection nanobodies, respectively. Both clones bind the APD-DD region but bind to distinctly different epitopes (Fig 2). Both capture and detection pairs were specific for TcdA in culture supernatants (Fig 3C and 3I, and Table 4) and able to detect TcdA from strains M7404, R20291, and VPI10463 (Fig 3D and 3J and Table 4). However, the A1D8/A1C3 pair was more sensitive, with a LOD of 0.019 ng/mL in buffer or culture supernatant, 0.075 ng/mL in resuspended feces, and 0.6 ng/mL in cecal content (Fig 3G, 3H, 3K and 3L). The A1D8/A1C3 pair was tested on supernatants from three cultures of M7404, R20291, and VPI10463 to reveal ranges of TcdA concentrations, 57–278 ng/mL, 264–323 ng/mL, and 376–1986 ng/mL, respectively, consistent with the ranges observed with the A2B10/A1A6 pair (Fig 3D and 3J and Table 4). The differences in sensitivity motivated us to test additional pairs, and to swap capture and detection nanobodies. Toxin titrations using capture/detection pair A2H9/A1C3 (Fig 3M) and A1C3/A2B10, A2B10/A1C3, A1D1/A1C3, and A2B5/A1C3 (S3 Fig) indicated that all pairs work *in vitro*. We then quantified TcdA in fecal and cecal samples from *C. difficile* infected mice two days after infection (Fig 3N, 3O and 3P). The A1D8/A1C3 pair was effective, while the A2H9/A1C3 and A2B10/A1A6 pairs were not. Using the A1D8/A1C3 pair, we measured a range of 7–189 pg TcdA per mg of feces across 10 mice and 204–1688 pg TcdA per mg of cecal contents in five mice (Fig 3O and 3P and Table 5). While one might expect to see higher concentrations of TcdA correlating with more significant weight loss in the animal, this was not observed at the two-day post-infection time point (S4 Fig). The higher concentration of TcdA in the cecum is consistent with prior studies showing the cecum as the major site of bacterial expansion and epithelial damage in the mouse model of CDI [12].

A similar approach was used to develop a sandwich ELISA for TcdB (Fig 4 and Table 6). TcdB sequences vary across strains and can be classified into 5 different sequence types with strains containing TcdB1 or TcdB2 being the most prevalent in human infection [8]. TcdB$_{VPI}$

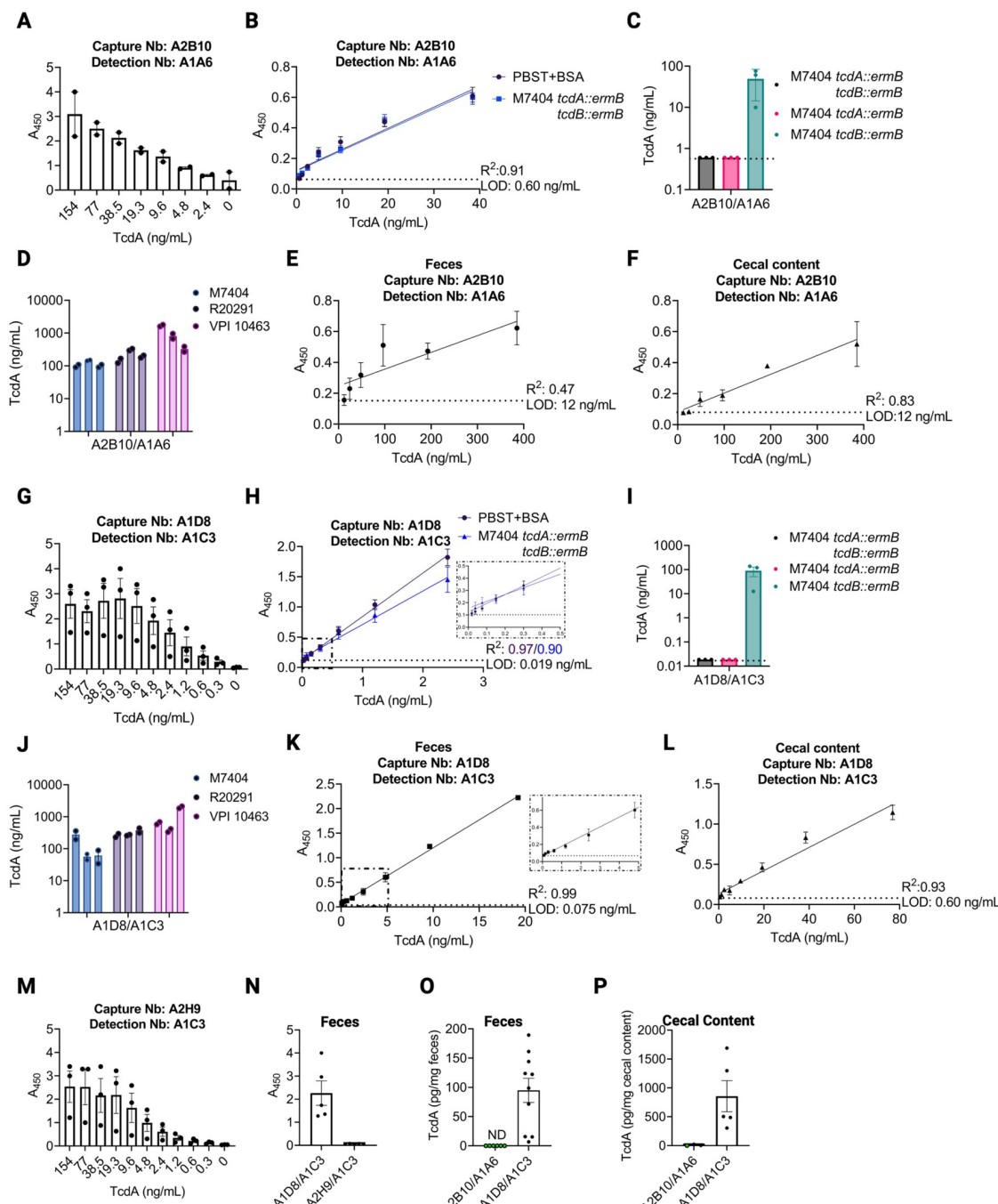

**Fig 3. Development of an anti-TcdA nanobody (Nb) based sandwich ELISA.** A). Detection of purified, recombinant TcdA (rTcdA) by sandwich ELISA using capture Nb A2B10 (anti-CROPs) and detection Nb A1A6 (anti-GTD) using two-fold serial dilutions of TcdA. B) A standard curve using two-fold serial dilutions of rTcdA in either PBST+BSA (black) or filtered supernatant of M7404 *tcdA*::*ermB* *tcdB*::*ermB* (blue). C) An ELISA measuring TcdA in filtered supernatant of M7404 *tcdA*::*ermB* *tcdB*::*ermB* (black), M7404 *tcdA*::*ermB* (pink), or M7404 *tcdB*::*ermB* (aqua). D) Use of the A2B10/A1A6 sandwich ELISA to quantify TcdA in *C. difficile* filtered supernatant of strains M7404 (light blue), R20291 (light purple), and VPI10463 (light pink). E) Standard curve for rTcdA in feces using the A2B10/A1A6 Nb pair and two-fold serial dilutions. Limit of detection (LOD) is noted by dashed line and was determined in roughly 50 mg/mL of feces. F) Standard curve for rTcdA in cecal content using the A2B10/A1A6 Nb pair and two-fold serial dilutions of TcdA. LOD noted by dashed line and was determined in roughly 500 mg/mL of cecal content. G) Evaluation of two anti-delivery domain (DD) Nbs (A1D8 and A1C3) in the sandwich ELISA assay using capture Nb A1D8 and detection Nb A1C3. A1D8 was used to coat the plate, followed by two-fold serial dilutions of rTcdA except where noted. H) A standard curve using two-fold serial dilutions of rTcdA in either PBST+BSA (black) or filtered supernatant of M7404 *tcdA*::*ermB*

*tcdB::ermB* (blue). I) An ELISA measuring TcdA in filtered supernatant of M7404 *tcdA::ermB tcdB::ermB* (black), M7404 *tcdA::ermB* M7404 (pink), M7404 *tcdB::ermB* (aqua). J) A1D8/A1C3 ELISAs recognize TcdA in *C. difficile* filtered supernatant of strains M7404 (light blue), R20291 (light purple), and VPI10463 (light pink). K) Standard curve for rTcdA in feces using A1D8/A1C3 Nb pair and two-fold serial dilutions of TcdA. LOD noted by dashed line and was determined in roughly 50 mg/mL of feces. Inset shows linear range down to 0.075 ng/mL of TcdA. L) Standard curve for rTcdA in cecal content using A1D8/A1C3 Nb pair and two-fold serial dilutions of TcdA. LOD noted by dashed line and was determined in roughly 500 mg/mL of cecal content. M) Evaluation of two anti-delivery domain nanobodies in the sandwich ELISA assay using capture Nb A2H9 and detection Nb A1C3. A2H9 was used to coat the plate, followed by two-fold serial dilutions of rTcdA except where noted. N) Evaluating two DD Nb ELISA's for detecting TcdA in the feces of *C. difficile* infected mice using a sandwich ELISA. ELISAs were performed using Nb combinations A1D8/A1C3 and A2H9/A1C3 in the feces of mice infected with *C. difficile* $TcdB_{GTX}$ two days post-infection. O) Using the fecal standard curves from E) and K), ELISAs were performed using Nb combinations A2B10/A1A6 and A1D8/A1C3 on feces from mice infected with *C. difficile* $TcdB_{GTX}$ two days post infection. Green filled circles represent samples that are below the LOD (dashed line). P) Using the cecal content standard curves from F) and L), ELISAs were performed using Nb combinations A2B10/A1A6 and A1D8/A1C3 in the cecal contents of mice infected with *C. difficile* $TcdB_{GTX}$ two days post-infection. Green filled circles represent samples that are below the LOD (dashed line). All ELISAs using purified protein were performed with technical duplicates and biological triplicates and error bars represent standard error of the mean (SEM). Measurements of native toxin within bacterial culture or infected animals represent the average of technical duplicates. Image created with BioRender.com license JK25IG3945.

was derived from a VPI10463 strain and is a member of the TcdB1 family of sequences, while $TcdB_{027}$ can be found in B1/NAP1/027 strains such as R20291 and M7404 and is a member of the TcdB2 family. A panel of capture/detection pairs were screened for their ability to detect both TcdB1 and TcdB2. Some recognized TcdB1 and TcdB2 equally well and some were effective against TcdB1 (VPI10463) but lacked sensitivity for TcdB2 ($TcdB_{027}$) (S5 Fig, panels C, E and F). We chose to pursue B2C11 and B0D10 as capture nanobodies, and B0E2 was biotinylated and used for detection. Both pairs were effective in recognizing TcdB1 and TcdB2 sequences (Fig 4A and 4G). The B2C11/B0E2 pair had a LOD of 2.1 ng/mL in buffer and culture supernatant while the B0D10/B0E2 pair had a LOD of 0.033 ng/mL in buffer and 0.65 ng/mL in culture supernatant (Fig 4B and 4H). Both pairs were specific for TcdB in culture supernatants (Fig 4C and 4I and Table 6) and could detect TcdB from strains M7404, R20291, and VPI10463 (Fig 4D and 4J and Table 6). Despite these similarities, the B2C11/B0E2 was moderately effective in detecting TcdB spiked into the feces and cecal contents of mice (Fig 4E and 4F), while the B0D10/B0E2 pair was completely unable to detect toxin in resuspended feces (Fig 4K). Therefore, the B2C11/B0E2 based ELISA was used to quantify TcdB in fecal and cecal samples from *C. difficile* infected mice two days after infection (Fig 4L). Since the severe diarrhea of mice at this time point typically limits the availability of cecal and fecal samples, mice were infected with an R20291 strain where the glucosyltransferase activity of TcdB was inactivated [12]. We measured a range of 198–862 pg TcdB per mg of feces in eight of the ten infected mice, with each mouse having a greater quantity of TcdB than TcdA (Table 5). There

**Table 4. Comparison of TcdA concentrations in bacterial supernatants between strains measured with different ELISA pairs.**

| Fig | Strain, Nb combination | Replicate (ng/mL) | | | Range (ng/mL) |
|---|---|---|---|---|---|
| | | 1 | 2 | 3 | |
| 3C | M7404 *tcdB::ermB*, A2B10/A1A6 | 74 | 12 | 78 | 12–78 |
| 3I | M7404 *tcdB::ermB*, A1D8/A1C3 | 138 | 13 | 120 | 13–138 |
| 3D | M7404, A2B10/A1A6 | 100 | 146 | 71 | 71–146 |
| 3J | M7404, A1D8/A1C3 | 62 | 278 | 57 | 57–278 |
| 3D | R20291, A2B10/A1A6 | 145 | 313 | 192 | 145–313 |
| 3J | R20291 A1D8/A1C3 | 268 | 378 | 264 | 264–378 |
| 3D | VPI10463, A2B10/A1A6 | 813 | 325 | 1690 | 325–1690 |
| 3J | VPI10463, A1D8/A1C3 | 633 | 376 | 1986 | 376–1986 |

**Table 5. Quantities of TcdA (Fig 3O and 3P) and TcdB (Fig 4L) detected in the cecal and fecal contents of mice two days post infection.**

| Mouse | TcdA pg per | | TcdB pg per | |
|---|---|---|---|---|
| | mg cecal content | mg feces | mg cecal content | mg feces |
| 1 | 1688 | 189 | 567 | 198 |
| 2 | 503 | 71 | 451 | 605 |
| 3 | 304 | 144 | 303 | 233 |
| 4 | 1297 | 99 | 1936 | 201 |
| 5 | 488 | 16 | 287 | 0 |
| 6 | | 122 | | 450 |
| 7 | | 161 | | 351 |
| 8 | | 124 | | 862 |
| 9 | | 7 | | 0 |
| 10 | | 15 | | 225 |
| Range | 304–1688 pg/mg | 7–189 pg/mg | 303–1936 pg/mg | 198–862 pg/mg |
| LOD | 12 pg/mg | 1.5 pg/mg | 106 pg/mg | 26 pg/mg |

were two mice, however, where the concentration of TcdB in feces fell below the limit of detection (26 pg TcdB/mg stool). We measured a range of 303–1936 pg TcdB per mg cecal content in five mice. These quantities were similar to the range of TcdA quantities (304–1688 pg), although there was no correlation in the TcdA and TcdB quantities when looking at individual mice (Table 5), and, as with TcdA, there was no correlation between TcdB toxin levels and weight loss at this time point (S4 Fig).

## Discussion

In this study, we used non-toxic point mutants of TcdA and TcdB without further inactivation, such as crosslinking or boiling, to immunize alpacas. We screened the nanobodies from TcdA- and TcdB-specific panning steps for binding to defined toxin domains, resulting in panels of domain-specific nanobodies. Prior studies have focused on the CROPS domain, in part because it was thought to contain the receptor-binding function and in part because it was thought to be immunodominant [9,14]. The data summarized in Table 1 suggest that no domain is immunodominant. This observation holds when considering both unique and total clones. A key difference between our study and the prior study where alpacas were immunized with full-length TcdA and TcdB mutants is that we did more rounds of immunization (8 versus ≤5) with shorter spacing between immunizations (2 weeks versus ≥3 weeks). These subtle differences may have promoted a more diverse immune response.

Despite broad domain coverage, both the TcdA and TcdB nanobody panels were composed of a relatively small number of antibody variable genes (Table 2). The dominant genes used IGHV 3–3 for TcdA and IGHV 3S53 for TcdB and are known to be among the most commonly used alpaca variable regions. However, they normally account for ~19% and ~15% of the total $V_{HH}$ repertoire [36], suggesting that a broad immune response in terms of epitopes targeted can arise from very restricted gene usage.

For our neutralization studies, we focused on nanobodies that bound outside the CROPS region, both because we had many such clones and because neutralizing CROPS-targeted nanobodies and antibodies have been extensively studied [26,37,38]. Despite this focus, we did find that TcdB CROPS-binding nanobodies were neutralizing. While the mechanism for this neutralization is currently unknown, it could suggest a role for CROPS-dependent glycan

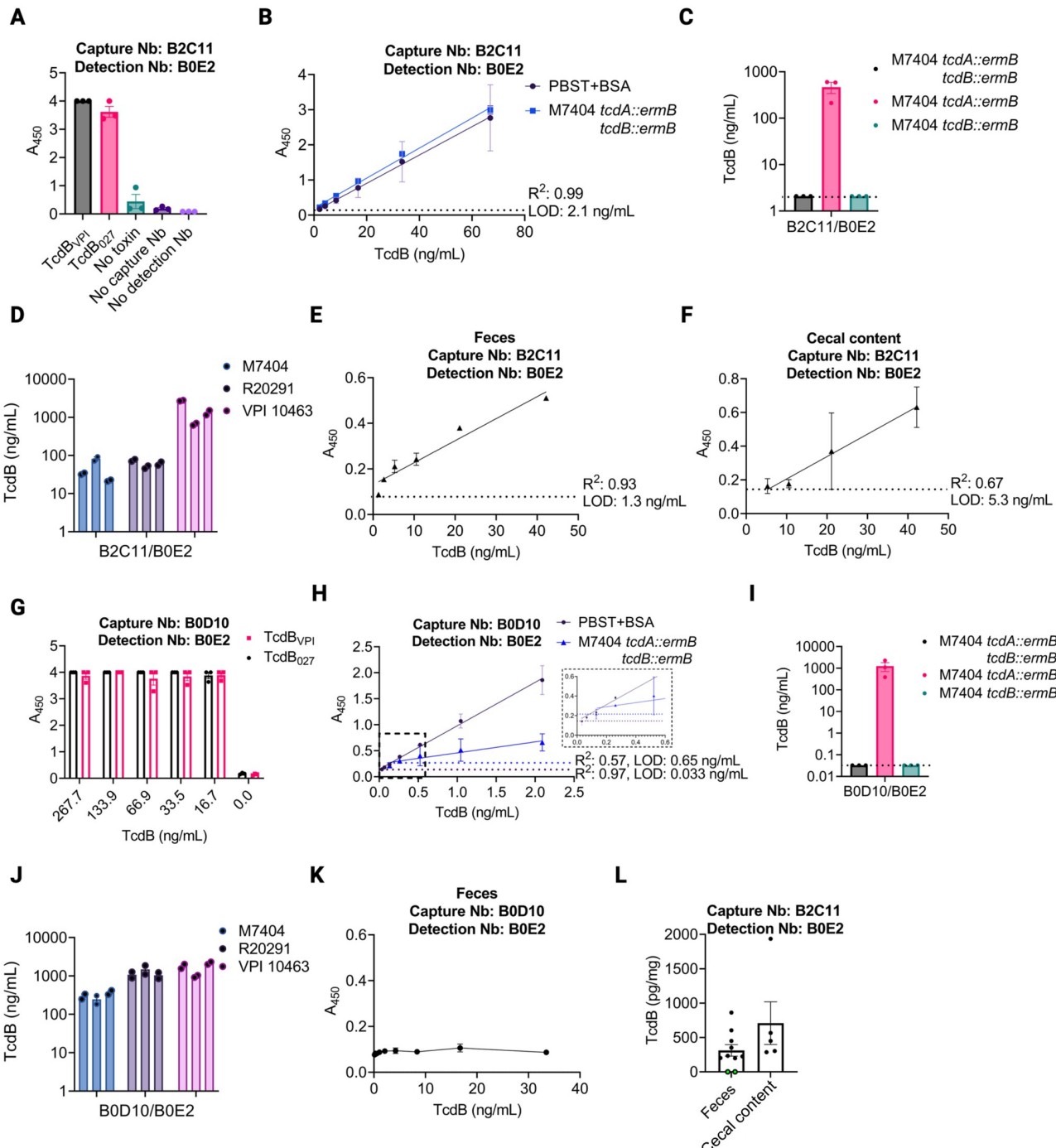

**Fig 4. Development of an anti-TcdB nanobody (Nb) based sandwich ELISA.** A). Detection of purified, recombinant TcdB (1 nM) from *C. difficile* strains VPI10463 or R20291 (rTcdB$_{VPI}$ or rTcdB$_{027}$, respectively) by sandwich ELISA using capture Nb B2C11 (anti-GTD) and detection Nb B0E2 (anti-DD). B) A standard curve using two-fold serial dilutions of rTcdB$_{VPI}$ in either PBST+BSA (black) or filtered supernatant of M7404 *tcdA::ermB tcdB::ermB* (blue). C) An ELISA measuring TcdB in filtered supernatant of M7404 *tcdA::ermB tcdB::ermB* (black), M7404 *tcdA::ermB* (pink), or M7404 *tcdB::ermB* (aqua). D) Use of the B2C11/B0E2 sandwich ELISA to quantify TcdB in *C. difficile* filtered supernatant of strains M7404 (light blue), R20291 (light purple), and VPI10463 (light pink). E) Standard curve for rTcdB$_{VPI}$ in feces using the B2C11/B0E2 Nb pair and two-fold serial dilutions of TcdB. Limit of detection (LOD) is noted by dashed line and was determined in roughly 50 mg/mL of feces. F) Standard curve for rTcdB$_{VPI}$ in cecal content using the B2C11/B0E2 Nb pair and two-fold serial dilutions of TcdB. LOD is noted by dashed line and was determined in roughly 500 mg/mL of cecal content. G) Evaluation of two anti-delivery domain (DD) Nbs (B0D10/B0E2) in the sandwich ELISA assay using capture Nb B0D10 and detection Nb B0E2. B0D10 was used to coat the plate, followed by two-fold serial dilutions of rTcdB$_{VPI}$ or rTcdB$_{027}$ except where noted. H) A standard curve using two-fold serial dilutions of rTcdB$_{VPI}$ in either PBST+BSA (black) or filtered supernatant of M7404 *tcdA::ermB tcdB::ermB* (blue). Inset shows data points below

0.5 ng/mL TcdB. I) An ELISA measuring TcdB in filtered supernatant of M7404 *tcdA::ermB tcdB::ermB* (black), M7404 *tcdA::ermB* M7404 (pink), M7404 *tcdB::ermB* (aqua). J) B0D10/B0E2 ELISAs recognize TcdB in *C. difficile* filtered supernatant of strains M7404 (light blue), R20291 (light purple), and VPI10463 (light pink). K) Standard curve for rTcdB$_{VPI}$ in feces using B0D10/B0E2 Nb pair and two-fold serial dilutions of TcdB. L) Using the fecal and cecal content standard curves from E) and F), ELISAs were performed using Nb combinations B2C11/B0E2 in the fecal and cecal contents of mice infected with *C. difficile* TcdB$_{GTX}$ two days post-infection. Green filled circles represent samples that are below the LOD (dashed line). All ELISAs using purified protein were performed with technical duplicates and biological triplicates and error bars represent standard error of the mean (SEM). Measurements of native toxin within bacterial culture or infected animals represent the average of technical duplicates. Image created with BioRender. com license LF25IG4OUD.

binding or as-yet unidentified protein receptors for TcdB. The DD, however, was the target of all potent TcdA binding clones as well as multiple potent TcdB binding clones. This was also unexpected, and a better understanding of the mechanism of inhibition used by these nanobodies merits further investigation. There is precedent, in the case of the monoclonal antibody PA41 for neutralizing antibodies to block GTD delivery. In that instance, PA41 binds GTD and blocks either pore formation or translocation of GTD through the endosomal pore, perhaps by blocking GTD unfolding [39]. Our interest in identifying neutralizers led us to speculate that nanobodies recognizing discrete epitopes on the surface of the DD might improve the efficacy of a sandwich ELISA assay. This is because the toxins are subjected to exogenous proteases within the complex milieu of the intestinal tract and GTD is known to be released by autoprocessing. Antibody reagents that bind to N- or C-terminal sequences of the toxins have the potential to lose efficacy if the toxins are proteolyzed. Indeed, we identified several nanobody pairs that were highly sensitive ELISA reagents in PBS or media, but ineffective in the feces or cecal contents of mice. Our effort to focus on nanobodies binding unique areas of the DD was an effort to mitigate that issue. For example, while CROPS-directed capture and GTD-directed detection (A2B10/A1A6) worked well for quantifying TcdA in buffer, it did not work well in feces or cecal content (Fig 3B, 3E and 3F). The switch to two nanobodies binding distinct epitopes on the TcdA DD, A1D8 and A1C3 (Fig 2), led to a marked improvement in our ability to detect TcdA in the cecal and fecal contents of mice (Fig 3K and 3L). However, adopting this strategy for TcdB did not result in the same improvements. B0D10 and B0E2, which target different sites on the TcdB DD (Fig 2), were a very efficient ELISA pair in buffer or culture supernatant but were completely unable to detect TcdB in feces (Fig 4). We speculate that this loss in efficacy results not from proteolytic degradation, but rather from occlusion or conformational disruption of the B0D10 nanobody epitope when the toxin is in a complex mixture. B2C11, a GTD targeted clone, can pair with B0E2 to efficiently measure toxin in feces despite having a higher LOD in buffer or media. If it is true that binding sites are being disrupted by a conformational change or occluded through carbohydrate, protein, or lipid binding, or other non-specific mechanisms such as aggregation, the process of identifying effective

**Table 6. Comparison of TcdB concentrations in bacterial supernatants between strains measured with different ELISA pairs.**

| Fig | Strain, Nb combination | Replicate (ng/mL) | | | Range (ng/mL) |
| --- | --- | --- | --- | --- | --- |
| | | 1 | 2 | 3 | |
| 4C | M7404 *tcdA::ermB*, B2C11/B0E2 | 212 | 587 | 506 | 212–587 |
| 4I | M7404 *tcdA::ermB*, B0D10/B0E2 | 389 | 1102 | 2285 | 389–2285 |
| 4D | M7404, B2C11/B0E2 | 33 | 22 | 82 | 22–82 |
| 4J | M7404, B0D10/B0E2 | 294 | 374 | 245 | 245–374 |
| 4D | R20291, B2C11/B0E2 | 75 | 49 | 62 | 49–75 |
| 4J | R20291, B0D10/B0E2 | 1079 | 1041 | 1486 | 1041–1486 |
| 4D | VPI10463, B2C11/B0E2 | 2730 | 1327 | 659 | 659–2730 |
| 4J | VPI10463, B0D10/B0E2 | 2167 | 1794 | 980 | 980–2167 |

targets for *in vivo* neutralization may need to also consider toxin epitope availability within the complex environment of the intestinal lumen.

In summary, we have developed a panel of nanobodies that bind specific structural and functional domains of TcdA and TcdB. In testing for neutralization, we noted that many of the potent neutralizers of TcdA bind epitopes within the delivery domain, a finding that could reflect roles of the delivery domain in receptor binding and/or the conserved role of pore-formation in the delivery of the toxin enzyme domains to the cytosol. We also identified potent neutralizers against TcdB that were targeting the GTD, DD, or CROPS domains. There are several factors that are expected to influence the potency of neutralization: epitope, binding affinity, and stoichiometry. Exploring the therapeutic potential of these reagents will require ongoing development. Nanobodies lack Fc functions and have short half-lives *in vivo*, but there is precedent for therapeutic and preventative development by expressing the nanobodies as chimeric fusions and through probiotic delivery strategies [40,41].

The nanobodies were also used for the creation of sandwich ELISA assays. While quantitation was possible by comparison with standard curves *in vitro*, we noted significant variation when comparing toxin levels in cultures grown from independent colonies or in mice infected from a common spore stock. This is not surprising when considering the phenotypic heterogeneity in *C. difficile* motility and toxin production which is maintained by recombination-mediated phase variation that is both reversible and stochastic [34]. There was also a loss in sensitivity when moving to heterogeneous cecal and fecal samples, a challenge that could reflect the complex nature of the sample, proteolytic degradation of the toxins, or some combination of the two. One way to circumvent these challenges would be to quantify the amount of active toxin in the cecal or fecal contents. While this is currently done using a Vero cell rounding assay and a reagent that neutralizes both TcdA and TcdB simultaneously, one may be able to develop methods for quantifying TcdA and TcdB independently using neutralizing nanobodies such as the ones presented in this study. For some strains, such as the R20291 strain used in this study, one will also need to neutralize the rounding associated with the CDT binary toxin.

In the conditions of this study, we observed significant variation in the quantities of TcdA and TcdB present within the cecal contents of mice at two days post infection (Table 5) and no correlation between toxin levels and weight loss (S4 Fig). This may suggest that differences in pathogenic responses caused by TcdA and TcdB are due to differences in activity and the host response rather than toxin levels. We note, however, that we used a strain where the glucosyltransferase activity of TcdB had been inactivated, the samples were evaluated at a single time point, and weight loss is an indirect readout for mechanistic study. An important priority for future studies will be to evaluate toxin levels at earlier timepoints and to identify alternate and/or more direct readouts of toxin activity that can be used to predict disease outcomes in the host. We plan to monitor the dynamics of TcdA and TcdB production over time and the impact of various experimental interventions on toxin production *in vivo*. We hope that these reagents can be of value to other researchers in need of similar assays.

## Material and methods

### Ethics statement

This study was approved by the Institutional Animal Care and Use Committee at Vanderbilt University Medical Center (VUMC) and performed using protocol M1700185-01. Immunization of Alpacas at Turkey Creek Biotechnology was done in accordance with Turkey Creek Biotechnology IACUC under protocol 18–01.

## Isolation of anti-TcdA and TcdB nanobodies

We developed TcdA and TcdB-targeted nanobodies in collaboration with Turkey Creek Biotechnology, Waverly, Tennessee, USA (in accordance with IACUC protocol 18–01) and the Vanderbilt Antibody and Protein Resource core facility (VAPR), Vanderbilt University, Nashville, Tennessee. The strategy was similar to other established protocols [42,43]. Two alpacas, one for each toxin, were immunized 8 times with 125 μg of purified mutant toxin mixed 1:1 by volume in Gerbu adjuvant FAMA (item 3030 Gerbu F from GERBU Biotechnik). Animals were immunized on days 0, 14, 29, 42, 55, 69, 90, and 104. Blood was drawn on days 42, 78, and 111 and similarly strong titers were seen at all timepoints. The TcdA used was an inactive mutant, a mutation to the DXD motif in the glucosyltransferase domain (D285N/D287N). The TcdB used was an L1106K mutant. Following immunization, blood was drawn into citrate containing blood bags, and PBMCs were isolated by centrifugation from ~100 mL of blood using SepMate centrifugal devices (STEMCELL Technologies). A cDNA library was made by reverse transcription using oligo dT primers and Superscript IV reverse transcriptase (ThermoScientific). A two-step, nested, PCR strategy was used to amplify coding regions of VHH fragments. This was done as described [42,43]. The resulting PCR fragments arise from B-cells of diverse lineages. These fragments were ligated into pBBR3, a modified pADL22 vector (Antibody Design Labs), containing sequences for in-frame, C-terminal, HA and hexahistadine tags. The plasmids were electroporated into high-efficiency TG1 cells (Lucent), and phage were produced using CM13 helper phage.

A single round of panning was done against 10 μg TcdAGTX or TcdB L1106K immobilized on MaxiSorp plates. Three wells of a MaxiSorp plate were coated overnight with 10 μg of toxin in PBS with three PBS coated wells serving as controls. Wells were blocked with 2% nonfat milk in PBS, washed, and incubated with $2 \times 10^{11}$ phage particles in blocking buffer for 1 hr. After extensive alternating washes with PBS and PBS + 0.5% Tween 20, phage were eluted with 100 μL 100 mM glycine pH 2.2, which was immediately neutralized with 1 M Tris-HCl pH 8. Recovered phage were used to infect TG1 E. coli, and single clones were picked into deep well 96 blocks of terrific broth with 100 μg/mL ampicillin. Plates were grown at 37˚C for 5 hrs followed by 28˚C overnight. Bacteria were pelleted and lysed by two freeze-thaw cycles with a total of 400 μL of PBS pH 7.4. Positive clones were identified with a modified ELISA using a MaxiSorp plate coated with 1 μg of neutravidin and 0.5 μg of biotinylated target per well, which was detected with 25–50 μL of periplasmic extract. The plate was developed using an anti-HA (12CA5) antibody followed by HRP-labeled goat anti-mouse secondary (Jackson ImmunoResearch) and 1 Step Ultra-ELISA TMB substrate (ThermoFisher).

## Nanobody expression & purification

Selected nanobody constructs were expressed and purified from SHuffle T7 E. coli (New England Biolabs). An overnight culture was diluted 1:100 into fresh Luria Broth with appropriate antibiotics and grown at 37˚C until a $OD_{600}$ of ~1.2 was reached. The temperature was lowered to 18˚C, 1 mM IPTG was added, and protein expression was continued overnight. Cells were harvested by centrifugation and lysed with an Emulsiflex. Nanobodies were purified from clarified supernatants using Talon affinity resin (Takara). Eluted nanobodies were concentrated in an Amicon centrifugal spin concentrator (3K MWCO), then run over an S-75 size exclusion column equilibrated in PBS pH 7.4. Fractions were analyzed by SDS-PAGE, pooled, snap-frozen in liquid $N_2$, and stored at -70 ˚C until use.

Selected nanobodies with C-terminal Avi-tags were biotinylated as described [44]. Briefly, a sample of Avi-tagged nanobody (100 μM) was incubated with 10 mM $MgCl_2$, 10 mM ATP, 1/$10^{th}$ mass BirA (Addgene plasmid 20857), and 50 μM D-biotin at 30 ˚C for 1 hr. Samples were

further purified over an S-75 size exclusion column equilibrated in PBS pH 7.4. Fractions containing biotinylated nanobody were analyzed by SDS-PAGE, and nanobody-containing fractions (>90% purity) were stored at -20 ˚C.

### *C. difficile* toxin constructs expression & purification

All recombinant TcdA and TcdB constructs were expressed in either *Bacillus megaterium* or *E. coli* and purified as described previously [39,45,46]. All plasmid constructs are listed in S2. Table.

### Bacterial growth conditions, medium, and strains

*C. difficile* strains were grown on BHIS (brain heart infusion-supplemented) medium or TY medium in a strict anaerobic environment within a COY anaerobic chamber (5% $H_2$, 5% $N_2$, and 90% $CO_2$). *E. coli* strains were maintained on Lysogeny Broth supplemented with respective antibiotics. All bacterial strains can be found in S3 Table.

### Toxin domain specificity assignment

Domain specificities for individual nanobody clones were determined by ELISA against isolated protein domains (TcdA-GTD, TcdA$_{1-1809}$ (APD-GTD-DD), TcdA-CROPs-R6R7, TcdB-GTD, TcdB$_{842-1834}$ (delivery domain), TcdB-CROPs). Briefly, protein domains were biotinylated and added to 96-well plates coated with NeutrAvidin (10 µg/mL solution). Bacterial supernatants from TG1 *E. coli* expressing the individual nanobodies were then added to the wells to test for binding against each antigen domain. Bound nanobodies were detected with a Li-Cor 800-labeled mouse anti-HA-tag antibody on a Li-Cor Odyssey imager.

### Nanobody sequence analysis

The DNA plasmids containing the nanobody clones were sequenced (Azenta). The corresponding amino acid sequences for the clones were manually analyzed to confirm the absence of frameshifts and spurious stop codons. Alignments of the full amino acid sequences were made in CLUSTAL OMEGA, and phylogenetic analysis was performed with RAxML using the raxmlGUI platform [47]. Results were visualized in ITOL v6.6 [48,49]. VHH germline gene analysis was performed with IMGT/HighV-QUEST against the *Vicugna pacos* (alpaca) reference directory, IGH gene locus [32].

### *In vitro* toxin neutralization assays

Multiple cell lines were used to evaluate neutralization of the toxins by individual nanobodies: T84 (TcdA), Caco-2 (TcdB), and Vero (TcdA, TcdB). Cells were plated in 96-well, black, clear-bottom, cell culture plates (Costar) at 3 x $10^3$ (T84, Caco-2) or 1.5 x $10^3$ (Vero) cells per well and incubated overnight at 37 ˚C and 5% $CO_2$. Purified toxins were incubated with serial dilutions of nanobodies for 0.5 hr at room temperature, then added to the cells. Plates were incubated for 72 hr, then the media were aspirated, and fresh media added to cells. CellTiter Blue (Promega) reagent (20 µL/well) was added, and the plates were incubated for 3.5 hr. Fluorescence was read in a Cytation plate reader (Bio-Tek) at 560 nm/590 nm excitation/emission. Cell viability was determined by subtraction of an untreated control and normalized to the toxin-only value. $EC_{50}$ was calculated in GraphPad Prism by least squares fit of the log(agonist) vs. response—variable slope (four parameters) model. For these experiments, $EC_{50}$ is the concentration of nanobody that increases viability to halfway between the zero nanobody baseline and maximum achievable protection.

## Negative stain electron microscopy

Purified recombinant $TcdA_{1-1832}$ or $TcdB_{1-1810}$ was run over a Superdex-200 size exclusion column equilibrated in 20 mM Tris, pH 8.0, 100 mM NaCl to remove potential aggregates. The toxins were mixed in a 1:2 molar ratio (100 nM toxin:200 nM nanobody) with individual purified nanobodies in the same buffer. The proteins were incubated at room temperature for 30 min then diluted four-fold in buffer immediately before application to the grids. Samples (3 μL) were applied to freshly glow-discharged, carbon-coated copper grids (Electron Microscopy Sciences), incubated for 1.5 min at room temperature, and stained for 1.5 min with freshly prepared 0.75% (mass/volume) uranyl formate [50]. Micrographs were collected at 62,000x magnification (1.7574 Å/pixel) with Serial EM software [51] on an FEI Tecnai F20 (200 keV) TEM equipped with a Gatan US4000 charge-coupled device camera. Individual particle datasets were picked for each nanobody complex, and two-dimensional alignment and classification was performed in RELION [52]. *Structural biology software used in this project was compiled and configured by* SBGrid [53].

## Anti-toxin ELISAs

ELISA plates (96-well flat-bottom; Nunc MaxiSorp) were coated overnight at 4 ˚C on an orbital shaker with 100 ng/mL solutions (in PBS) of either A2B10, A1D8 (anti-TcdA) or B2C11, B0D10 (anti-TcdB) capture nanobody. Next, the plates were washed 4 times with PBS + 0.05% Tween-20 (PBS-T), and incubated with blocking buffer (PBS-T + 2% BSA) for 2 hr at room temperature with shaking, followed by 4 washes with PBS-T. For the toxin standard curves, each toxin was diluted to 1 nM (308 or 270 ng/mL of TcdA/B) in blocking buffer and a 2-fold dilution series was set up. The toxins were added to the coated plate, incubated for 1 hr at room temperature with shaking, then washed 4 times with PBS-T. For the supernatant standard curves, clarified bacterial supernatant was serially diluted (2-fold) and added to the plates. The plates were then incubated for 2 hours at room temperature with shaking and washed 4 times with PBS-T. Captured toxins were detected by addition of 100 μL of solutions containing biotinylated Avi-tagged detection nanobodies: A1A6 (20 ng/mL), A1C3 (5 ng/mL), and B0E2 (5 ng/mL). Plates were incubated for 2 hr at room temperature with shaking and washed 4 times with PBS-T. HRP conjugated Streptavidin (ThermoScientific) was diluted 1:20,000 in blocking buffer, added to the plates, and incubated for 1 hr at room temperature with shaking. Plates were washed five times with PBS-T, then 75 μl of 1 Step UltraTMB ELISA substrate solution (equilibrated to room temperature) was added to each well. ELISAs were quenched with $H_2SO_4$ and read at 450 nM in a Cytation plate reader (Biotek). To calculate the total amount of toxin captured from the samples, all plates contained a full standard curve. The limit of detection (LOD) was defined as the lowest concentration within the linear signal range that could be distinguished from the no-toxin control.

## Toxin secretion in *C. difficile*

All strains were streaked onto BHIS with thiamphenicol plates. Well-isolated colonies were picked into TY medium and grown for 16 hr under anaerobic conditions (without shaking). The following day, the resulting growth was sub-cultured 1:200 into fresh TY medium and grown for 24 hr. The cultures were centrifuged at 3000xg for 5 min, then the supernatant was filtered through a 0.8 μM syringe filter and stored at -70 ˚C until use.

For anti-toxin ELISAs, the supernatants were thawed on ice and serially diluted 2-fold in PBS-T + 2% BSA. To account for plate-to-plate variability, rTcdA or rTcdB standard curves were included on each plate, and the total toxin was calculated based on those standard curves.

Concentrations of toxin that fell at or below the limit of quantification are plotted on the curve as such.

## Measurement of toxin concentrations in a murine model

Mice were monitored daily and humanely euthanized via $CO_2$ asphyxiation at various time points. All animal experiments were performed using the cefoperazone mouse infection model using $10^4$ spores of *C. difficile* [54].

TcdA and TcdB standard curves were generated using mice infected with R20291 *tcdA*::*CT* *tcdB*::*CT*. Stool was collected daily from mice, and cecal content was harvested at four days post infection. Samples were frozen in liquid nitrogen and stored at -80˚C. Stool and cecal content was homogenized in PBS to a concentration of roughly 50 mg/mL or 500 mg/mL, respectively. Unlike samples secreted into TY media, cecal and fecal samples could not be reliably filtered. ELISAs were performed as described above, except the toxins and fecal or cecal content mixture was incubated at 37˚C for 30 min with shaking followed by 1.5 hrs at room temperature to simulate conditions within the mouse. To account for plate-to-plate variability, rTcdA or rTcdB standard curves were included on each plate.

TcdA and TcdB were measured in mice infected with a mildly attenuated strain, R20291 TcdB$_{GTX}$, as it is difficult to obtain fecal samples from mice with the severe diarrhea associated with the wild-type strain [12]. Stool and cecal content were collected at two days post infection, and toxin concentrations were determined on the same day. Briefly, fecal and cecal content were resuspended in PBS to a concentration of roughly 50 mg/mL or 500 mg/mL, respectively. A sample of the fecal slurry was plated for CFU's on TCCFA (10% Taurocholate, 10 mg/mL D-cycloserine, 10 mg/mL cefoxitin, 10 mg/mL fructose agar) semi-selective medium to ensure colonization. The remainder of the slurries were subjected to an ELISA as described above. Briefly, slurries were serially diluted 2-fold in PBS-T + 2% BSA and added to ELISA plates. To account for plate-to-plate variability, rTcdA or rTcdB standard curves were included on each plate.

## Supporting information

**S1 Fig. Cladograms representing amino acid sequence analysis and domain specificity of TcdA (*A*) and TcdB (*B*) nanobodies.** Highly identical clones from each panel were removed (TcdA cutoff 100% ID, TcdB cutoff 95% ID) for analysis. Total clone numbers for each group are in parentheses. Nanobodies used in the experiments in this study are highlighted in *yellow*. Domain specificity was determined by ELISA (*red circle* GTD, *blue square* DD or APD-DD, *green star* CROPs).
(DOCX)

**S2 Fig. Data graphs of toxin neutralization assays for TcdA (top) or TcdB (bottom).** Analysis was performed in GraphPad Prism by least squares fit of the model: log(agonist) vs. response—variable slope (four parameters). Error bars represent standard error for triplicate experiments, where possible.
(DOCX)

**S3 Fig. Screening nanobody pairs for anti-TcdA ELISAs.** Detection of purified, recombinant TcdA by sandwich ELISA using: A) capture Nb A1C3 (anti-DD) and detection Nb A2B10 (anti-CROPs), B) capture Nb A2B10 (anti-CROPs) and detection Nb A1C3 (anti-DD), C) capture Nb A1D1 (anti-DD) and detection Nb A1C3 (anti-DD), and D) capture Nb A2B5 (anti-DD) and detection Nb A1C3 (anti-DD). Two-fold serial dilutions of rTcdA were used, except where noted. All ELISAs were performed in biological triplicate and error bars represent

standard error of the mean (SEM). Image created with Biorender.com license number PU25IG5KBX.
(DOCX)

**S4 Fig. Comparison of toxin concentrations and weight loss in a murine model of *C. difficile* infection.** Mice were infected with $10^4$ spores of R20291 TcdB$_{GTX}$, sacrificed after 2 days, weighed, and cecal contents and fecal contents were collected for TcdA/TcdB quantification. A, B) Individual mouse fecal or D, E) cecal toxin quantities from Figs 3 and 4 were plotted against weight loss (expressed as a percent of starting weight). In addition, individual C) fecal or F) cecal TcdA quantities were plotted against the corresponding TcdB quantities. Image created with Biorender.com license number AK25IG337P.
(DOCX)

**S5 Fig. Screening nanobody pairs for anti-TcdB ELISAs.** Detection of purified, recombinant TcdB (1 nM) from *C. difficile* VPI10463 (a TcdB1 strain) or R20291 (a TcdB2 strain), labeled rTcdB$_{VPI}$ or rTcdB$_{027}$, by sandwich ELISA using: A) capture Nb B2C11 (anti-GTD) and detection Nb B1A11 (anti-CROPs), B) capture Nb B2F11 (anti-CROPs) and detection Nb B0E2 (anti-DD), C) capture Nb B2F11 (anti-CROPs) and detection Nb B1A11 (anti-CROPs), and D) capture Nb B2F11 (anti-CROPs) and detection Nb B0B11 (anti-GTD). Evaluation of anti-DD Nbs E) B0A12/B0E2 and F) B1C11/B0E2 in the sandwich ELISA assay using capture Nb B0A12 or B1C11, respectively, and detection Nb B0E2. B0A12 or B1C11 was used to coat the plate, followed by two-fold serial dilutions of rTcdB$_{VPI}$ or rTcdB$_{027}$ except where noted. All ELISAs were performed in biological triplicate and error bars represent standard error of the mean (SEM). Image created with Biorender.com license number CQ25IG5GXK.
(DOCX)

**S1 Table. Sequences of nanobodies used in this study.**
(DOCX)

**S2 Table. Plasmids used in this study.**
(DOCX)

**S3 Table. Strains used in this study.**
(DOCX)

## Acknowledgments

We would like to thank Shannon Dorr for her assistance in nanobody purification, Stacey Seeback and the VAPR core for nanobody isolation, sequencing, and domain screening, and Anna Smith for harvesting fecal and cecal material from *C. difficile* infected mice. EM data collection was conducted at the Center for Structural Biology Cryo-EM Facility at Vanderbilt University.

## Author Contributions

**Conceptualization:** Shannon L. Kordus, Heather K. Kroh, Brian E. Wadzinski, D. Borden Lacy, Benjamin W. Spiller.

**Data curation:** Shannon L. Kordus, Heather K. Kroh, F. Christopher Peritore-Galve, John A. Shupe, D. Borden Lacy.

**Formal analysis:** Shannon L. Kordus, Heather K. Kroh, Brian E. Wadzinski, D. Borden Lacy, Benjamin W. Spiller.

**Funding acquisition:** D. Borden Lacy, Benjamin W. Spiller.

**Investigation:** Shannon L. Kordus, Heather K. Kroh, Rubén Cano Rodríguez, Rebecca A. Shrem, F. Christopher Peritore-Galve, John A. Shupe, Brian E. Wadzinski, Benjamin W. Spiller.

**Methodology:** Shannon L. Kordus, Heather K. Kroh, Rebecca A. Shrem, F. Christopher Peritore-Galve, D. Borden Lacy, Benjamin W. Spiller.

**Project administration:** John A. Shupe, D. Borden Lacy, Benjamin W. Spiller.

**Resources:** D. Borden Lacy, Benjamin W. Spiller.

**Supervision:** Brian E. Wadzinski, D. Borden Lacy, Benjamin W. Spiller.

**Writing – original draft:** Shannon L. Kordus, Heather K. Kroh, D. Borden Lacy, Benjamin W. Spiller.

**Writing – review & editing:** Shannon L. Kordus, Heather K. Kroh, Rubén Cano Rodríguez, Rebecca A. Shrem, F. Christopher Peritore-Galve, John A. Shupe, Brian E. Wadzinski, D. Borden Lacy, Benjamin W. Spiller.

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
