## [Decision Letter · Decision Letter 0]

29 Jul 2023

Dear Dr. Spiller,

Thank you very much for submitting your manuscript "Nanobodies against C. difficile TcdA and TcdB reveal unexpected neutralizing epitopes and provide a toolkit for toxin quantitation in vivo" for consideration at PLOS Pathogens. As with all papers reviewed by the journal, your manuscript was reviewed by members of the editorial board and by several independent reviewers. The reviewers appreciated the attention to an important topic. Based on the reviews, we are likely to accept this manuscript for publication, providing that you modify the manuscript according to the review recommendations.

The reviewers were all enthusiastic about the significance and quality of your work and its presentation in the manuscript. A number of minor points were raised that should be addressed, most of which involve clarification of experimental details and the nanobodies. 

Sincerely,

Nicole M Koropatkin, Ph.D.

Academic Editor

PLOS Pathogens

Michael Otto

Section Editor

PLOS Pathogens

Kasturi Haldar

Editor-in-Chief

PLOS Pathogens

orcid.org/0000-0001-5065-158X

Michael Malim

Editor-in-Chief

PLOS Pathogens

orcid.org/0000-0002-7699-2064

Reviewer Comments (if any, and for reference):

Reviewer's Responses to Questions

**Part I - Summary**

Reviewer #1: In this study, Spiller and colleagues produced mutants of Clostridioides difficile toxins A and B (TcdA and TcdB) that lacked cell entry / intoxication capability or target protein glucosylation activity and then used them as immunogens. The immunogens were then used to repeatedly immunize alpacas such that they produced a high anti-toxin nanobody reponse. PBMCs were isolated from blood samples and sequencing, expression, and phage display used to determine the properties of the recombined nanobody gene sequences and to produce nanobodies for functional testing. A variety of assays were performed to epitope map binding to toxin and to determine if the nanobodies could neutralize toxin and/or detect it in different environments.

Overall, this is a well-designed study coupled to a scholarly presentation, that in this reviewers requires some clarifications for the readership. The work is significant in the C. difficile field for the following reasons: (1) recent vaccine failures suggest the need for testing engineered immunogens; (2) there are very few therapeutics available for treatment; (3) basic understanding of the immunogenicity of toxin-based antigens is incomplete (4) reagents and tools for research need to be improved and made available to the field. This study addresses each of these points of significance.

Reviewer #2: This manuscript describes the generation and characterization of ~200 unique alpaca-derived nanobodies that target C. difficile TcdA or TcdB. This is a technology that has previously been described by other groups. However, here, the investigators have identified Nb that can neutralize toxin activity via inhibition of additional domains of the toxins. The investigators cleverly use negative staining EM to identify domains of TcdA and TcdB targeted by individual Nbs and proceed to develop a sandwich ELISA assay to detect TcdA and TcdB in the feces and cecal content of infected mice. These Nbs are likely to be of use to numerous investigators in downstream applications.

Reviewer #3: The clearly written manuscript by Kordus et al. describes the development and characterization of nanobodies that specifically distinguish between TcdA and TcdB toxins. Not only do the authors identify the regions of the toxins bound by different nanobodies and their ability to neutralize toxin activity, they also develop a highly sensitive ELISA assay to quantify toxin levels even in complex mixtures like cecal and fecal contents at pg levels. Unlike existing widely used methods, their assay can distinguish between the TcdA and TcdB toxins unlike the most widely used methods, and the ELISA assay is far less labor intensive. Thus, the authors have developed an assay that represents a significant improvement over existing methods. The assay also enables researchers to ask questions about how individual toxin levels vary over the length of an infection and correlate these measurements with disease severity.

Related to this point, the authors find that similar amounts of TcdA and TcdB toxin are detected in cecal and fecal contents of mice, suggesting that the differences observed in the pathogenicity caused by TcdA vs. TcdB are due to differences in activity/ host cell targeting rather than levels per se (some reports suggest that TcdB levels might exceed TcdA, but they show that this is not the case). The authors may want to specifically comment on this finding in their Discussion section.

**Part II – Major Issues: Key Experiments Required for Acceptance**

Reviewer #1: No major concerns.

Reviewer #2: 1. Is anything known about the absolute or relative binding affinities of the different HA-tagged Nbs to the toxins? Might differences in neutralization properties reflect differences in binding affinity? Seems that this information is important particularly as the investigators are proposing that others use these nanobodies as tools in their research. Furthermore, might differences in binding affinities explain the sensitivity of the sandwich ELISAs conducted with different pairs of Nbs? As well as differences in their ability to neutralize? For example, can one conclude that the two toxins are differentially neutralized by Nbs that bind the CROPS domains of the two toxins, if it is not known whether those Nbs being compared bind to the CROPS domains with similar binding affinities?

2. In the data availability statement, it says that “Sequences to clones developed in this work can be found in the supplemental figures”. Presumably. this sentence is referring to the sequences of the nanobodies. However, I was unable to find this information anywhere in the supplemental materials. Given that the focus of this manuscript is the characterization of the anti-TcdA and anti-TcdB nanobodies and their potential to be used by others in future studies, a supplemental table that contains the unique sequences of all the 69 anti-TcdA and 126 anti-TcdB nanobodies isolated should be added. In such a table, it would be helpful to demarcate which of the sequences of the nanobodies map to their V, D, J and CDR3 regions.

3. Is detection of toxins in the feces and/or cecal contents improved when fecal material is filtered?

Reviewer #3: (No Response)

**Part III – Minor Issues: Editorial and Data Presentation Modifications**

Reviewer #1: Some clarifications throughout would be helpful to readers. Specifically:

(1) As was mentioned in discussion, please modify methods section to provide a clear timeline for alpaca immunizations and bleeds. A clearer rationale for the number of immunizations and their timing, and any information on tracking titers would be helpful.

(2) Please provide more information on Gerbu adjuvant (formulation, composition, supplier).

(3) Please comment on what B cells are being sequenced in PBMCs from immunized alpacas. Given that 8 bleeds have taken place and that they were spaced 2-3 weeks apart, presumably there are toxin-specific activated Bcells and memory B cells, as well as plasmablasts, and plasma cells.

(4) A sentence of two for the benefit of non-specialists to better understand negative staining electron microscopy would be helpful.

(5) The clonality of the alpaca nanobody response to each toxin is clearly described but its relevance to human responses to those toxins is not clear. Given, the variety of B lineage types in the PBMCs, it should be acknowledged that the data do not reveal the clonality of specific lineages (memory versus plasma cell for example).

(6) Can the authors discuss whether the difference in nanobody recognition of different epitopes on TcdA versus TcdB extrapolate to other species? Since Ab recognition in mouse and human seem to concentrate in the CROP regions of TcdB, albeit not exclusively in that region, are similar differences observed between responses to TcdA and TcdB in these species?

Reviewer #2: 1. It would be useful for the reader to divide results section into subsections with headers.

2. Line 173: Are the number of potential V, D and J regions of heavy chain antibodies known? If so, might provide some context to provide some background information in this area.

3. Line 180: Would provide more background and clarify that the complementarity determining regions are those that bind the antigen.

4. Line 186/188. Would consider adding labeling the two clades in sFig. 1 A and B as referred to in the text. In the figure, they are referred to as top and bottom panels.

5. Lines 195, as both cell lines were not used with each nanobody, would change and to and/or. Please include information and references regarding the TcdA and TcdB receptors known to be present on Vero, Caco-2 and T84 cells. What does it mean when the EC50 is vastly different between cell types, i.e., with A1G4? Would also define EC50 somewhere in the text.

6. The text in Figures S3 and S4 is not legible and unable to evaluate this data. Text in Fig. S5 is very pigmented when blown up. Were these figures really made using Biorender? Doesn’t seem like right software for this.

7. Line 253-254. I think the investigators are referring to Figure S3 not Figure S2.

8. Reference to Fig. S5 precedes Fig. S4.

9. Line 268-269: Why focus on developing an assay to measure TcdB1 and TcdB2 as opposed to all five TcdB subtypes? Is it feasible that in future studies a standardize clinical sandwich ELISA could be developed to diagnose all five toxin subfamilies? Thoughts reflecting this possibility would be interesting to include in the discussion.

Reviewer #3: The authors find that there is no correlation between the levels of toxins measured and the weight loss observed in individual mice. This could suggest that there are additional factors that drive disease severity beyond the toxins (e.g. host responses). Since the authors point out in their discussion that the ELISA assay does not distinguish between whether toxins are intact and functional, it would be helpful to determine if the toxin levels they measure correlate with bulk toxin activity (TcdA + TcdB combined) using the standard Vero cell assay. This information would allow them to demonstrate that their ELISA assay is a good proxy for toxin levels and activity in in vivo samples. Aside from this experiment, which is not essential for publication, I have minor suggestions on data presentation and the Discussion points.

Related to the earlier point, it would be helpful to move Supplemental Figure 5 to the main paper, since an important contribution of the paper is their systematic demonstration that toxin levels do not necessarily correlate with disease symptoms.

Do the authors think that they can identify conserved epitopes with their approach that will broadly neutralize or identify the diverse toxin types, especially for TcdB? Their data currently suggest that the nanobodies against TcdB are not able to recognize different toxinotypes.

Why did the authors pursue a different immunization strategy for TcdA vs. TcdB (different mutations were used to immunize llamas.)? It would be helpful to provide this information in the manuscript, since the authors have previously shown that TcdB is more conformationally flexible than TcdA. It would also be helpful to comment on why they observe differences in the neutralization activity of their nanobodies for TcdA vs. TcdB in terms of where the nanobodies target and how that can lead to toxin neutralization.

Why did the authors use TcdB2 GTD instead of TcdB1?

What was the rationale for choosing an antibody as a capture vs. a detection nanobody?

I would suggest adding a final model figure that shows where the most potent antibodies bind and how the antibodies performed in the ELISA assay to summarize their findings in a more visual manner to highlight the differences in the TcdA and TcdB nanobody inhibitory function that are summarized in the Discussion section.

The authors might want to consider revising Table 1 so that it is incorporated into part of Figure 2 as a heat map rather than a table.

Line 91: consider changing “and” to “but”

Line 95: consider adding “repetitive” before “sequence blocks”

Line 256: please add a comma before “while”

Line 297: please hyphenate “domain-specific”

Line 318: the comma after “or” can be deleted; consider revising the sentence so that it reads “… role for CROPS-dependent glycan binding or as-yet-unidentified protein receptors for TcdB.”

Line 323: this sentence was a little hard to parse. Consider “However, in this instance, binding of PA41 to the GTD blocks pore formation or the ability of the GTD to be transited through the endosomal pore.”

Line 328: consider replacing “In addition to autoprocessing and release of the GTD” with “This is because”

Line 332: the comma on this line can be deleted

Line 347: consider revising the sentence so that it reads “If it is true that binding sites are being disrupted…also consider toxin epitope availability within the complex environment of the intestinal lumen” (add “it is” and delete “of” before “environment”)

Line 370: “the” can be deleted

Line 371: consider replacing “complex” with “heterogeneous nature of these samples”

Line 307: please define what “variable gene usage” is

PLOS authors have the option to publish the peer review history of their article (what does this mean?). If published, this will include your full peer review and any attached files.

Reviewer #1: No

Reviewer #2: No

Reviewer #3: No

Figure Files:

Data Requirements:

Reproducibility:

References:

---

## [Decision Letter · Decision Letter 1]

7 Oct 2023

Dear Dr. Spiller,

We are pleased to inform you that your manuscript 'Nanobodies against C. difficile TcdA and TcdB reveal unexpected neutralizing epitopes and provide a toolkit for toxin quantitation in vivo' has been provisionally accepted for publication in PLOS Pathogens.

Best regards,

Nicole M Koropatkin, Ph.D.

Academic Editor

PLOS Pathogens

Michael Otto

Section Editor

PLOS Pathogens

Kasturi Haldar

Editor-in-Chief

PLOS Pathogens

orcid.org/0000-0001-5065-158X

Michael Malim

Editor-in-Chief

PLOS Pathogens

orcid.org/0000-0002-7699-2064

Reviewer Comments (if any, and for reference):

Reviewer's Responses to Questions

**Part I - Summary**

Reviewer #1: Authors have been responsive to critiques and satisfactorily answered my previous queries.

Reviewer #2: The authors have address all of my concerns.

Reviewer #3: The authors have done a thorough job responding to the reviews. The wording changes improve an already excellent manuscript. Not a critical point, but my suggestion for the heat map for Table 1 was to incorporate some shading to the table (a feature available in excel), so that it would be visually easy to see based on color intensity the relative specificity of the nanobodies.

**Part II – Major Issues: Key Experiments Required for Acceptance**

Reviewer #1: (No Response)

Reviewer #2: (No Response)

Reviewer #3: n/a

**Part III – Minor Issues: Editorial and Data Presentation Modifications**

Reviewer #1: (No Response)

Reviewer #2: (No Response)

Reviewer #3: n/a

PLOS authors have the option to publish the peer review history of their article (what does this mean?). If published, this will include your full peer review and any attached files.

Reviewer #1: No

Reviewer #2: No

Reviewer #3: No

---

## [Editor Report · Acceptance letter]

18 Oct 2023

Dear Dr. Spiller,

We are delighted to inform you that your manuscript, "Nanobodies against C. difficile TcdA and TcdB reveal unexpected neutralizing epitopes and provide a toolkit for toxin quantitation in vivo," has been formally accepted for publication in PLOS Pathogens.

Best regards,

Kasturi Haldar

Editor-in-Chief

PLOS Pathogens

orcid.org/0000-0001-5065-158X

Michael Malim

Editor-in-Chief

PLOS Pathogens

orcid.org/0000-0002-7699-2064